# Beyond Antagonism: The Interaction Between *Candida* Species and *Pseudomonas aeruginosa*

**DOI:** 10.3390/jof5020034

**Published:** 2019-04-19

**Authors:** Ruan Fourie, Carolina H. Pohl

**Affiliations:** Department of Microbial, Biochemical and Food Biotechnology, University of the Free State, Bloemfontein 9301, South Africa; fourier2@icloud.com

**Keywords:** *Candida albicans*, interaction, *Pseudomonas aeruginosa*

## Abstract

There are many examples of the interaction between prokaryotes and eukaryotes. One such example is the polymicrobial colonization/infection by the various opportunistic pathogenic yeasts belonging to the genus *Candida* and the ubiquitous bacterium, *Pseudomonas aeruginosa*. Although this interaction has simplistically been characterized as antagonistic to the yeast, this review highlights the complexity of the interaction with various factors influencing both microbes. The first section deals with the interactions *in vitro*, looking specifically at the role of cell wall components, quorum sensing molecules, phenazines, fatty acid metabolites and competition for iron in the interaction. The second part of this review places all these interactions in the context of various infection or colonization sites, i.e., lungs, wounds, and the gastrointestinal tract. Here we see that the role of the host, as well as the methodology used to establish co-infection, are important factors, influencing the outcome of the disease. Suggested future perspectives for the study of this interaction include determining the influence of newly identified participants of the QS network of *P. aeruginosa*, oxylipin production by both species, as well as the genetic and phenotypic plasticity of these microbes, on the interaction and outcome of co-infection.

## 1. Introduction

Nature provides many examples of the interaction between prokaryotes and eukaryotes [1,2]. These interactions can have various outcomes and may provide an advantage to one party (e.g., through antagonism of the other) or both parties (mutualism). Interactions between fungi and bacteria are abundant in many habitats, including the human body [3,4,5]. The identification of these interactions, especially in the human host, have been performed through culture-dependent techniques or culture-independent techniques, the latter being able to identify the presence of microbial intruders at low cell counts that could play a large role in infection. One of the most well-documented bodily sites where polymicrobial interaction takes place is the gastrointestinal tract, colonized by a plethora of microbial species [6]. Furthermore, polymicrobial infections occur frequently in the urinary tract, wounds, where it can negatively affect wound healing, as well as in the lungs, especially in patients predisposed to infection by altered immunity, as in the case with cystic fibrosis (CF). The competition posed by close proximity and nutrient limitation can result in altered metabolism and virulence factor expression as well as altered host immunity that may lead to a competitive advantage to species within a polymicrobial infection [6,7]. This interaction between fungi and bacteria can cause a considerable increase in morbidity and mortality compared to single species infection [8,9].

One of the principle models for studying the interaction between eukaryotic and prokaryotic interaction in the context of human health is the association between *Candida albicans*, a polymorphic fungus, and the ubiquitous bacterium *Pseudomonas aeruginosa*. The study of this interaction dates back to the 1970s, when several authors observed that *P. aeruginosa* had an inhibitory effect on the growth of *C. albicans* [10,11]. This phenomenon is not confined to these two species. Kerr (1994) found that *Pseudomonas cepacia* also has this inhibitory effect, while other potentially pathogenic *Candida* spp. (*C. dubliniensis*, *C. glabrata*, *C. guillermondii*, *C. kefyr*, *C. krusei*, *C. lusitaniae*, *C. parapsilosis*, *C. pseudotropicalis and C. tropicalis*) can also be susceptible to inhibition by *Pseudomonas* [12,13,14]. Today we know that many factors can mediate the interaction between these organisms, with the best-studied example being the interaction between *P. aeruginosa* and *C. albicans*.

## 2. *In Vitro* Interaction between *Pseudomonas aeruginosa* and *Candida* Species

### 2.1. Role of Cell Wall Components in the Interaction

Hogan and Kolter (2002) showed that *P. aeruginosa* preferentially attaches to *C. albicans* hyphae, effectively killing them, probably to gain nutrients for growth [1]. This was not the case for *C. albicans* yeast cells. In addition, they found that *P. aeruginosa* mutants lacking type IV pili, could attach to *C. albicans* hyphae, but were unable to kill them. They speculated that the role of pili might involve pilus retraction to bring the bacterium into close contact with the hyphae or that the pili act as sensors, signaling attachment to the fungal surface. Other cell wall-associated compounds of both *P. aeruginosa* and *C. albicans* are also important in this interaction. Bacterial lipopolysaccharide (LPS), including *P. aeruginosa* LPS, has adverse effects on *C. albicans* biofilms [14,15]. These include a decrease in filamentation, biofilm metabolic activity (including glycolysis) and growth. In contrast, peptidoglycan triggers filamentation in *C. albicans* [16]. Brand and co-workers (2008) investigated the role of *C. albicans* hyphal-specific cell wall proteins and mannoproteins in the selective attachment and killing of hyphae, and found that neither the major hyphal-specific proteins (Hyr1p, Hwp1p and Als3p), nor enzymes involved in *N*-glycosylation of cell wall proteins played a role, but that O-mannan probably protects against *P. aeruginosa* killing activity [17]. Whereas the bulk of research has been done on the interaction between *C. albicans* and *P. aeruginosa*, less is known regarding the interaction of *P. aeruginosa* with other *Candida* species. However, Bandara and co-workers (2010) did indicate a mutually suppressive interaction between *P. aeruginosa* and five non-albicans *Candida* species, i.e., *C. glabrata*, *C. tropicalis*, *C. parapsilosis*, *C. dubliniensis* and *C. krusei* in an in vitro dual species biofilm model [14]. They observed species-specific variations of inhibition of *Candida* biofilms, with almost complete inhibition of *C. albicans*, *C. glabrata* and *C. tropicalis* biofilms after 48 hours. Interestingly, although research from Brand and co-workers (2008) indicated selective attachment and killing of hyphal cells, Bandara and co-workers (2010) observed attachment of bacterial cells to blastospores, indicating that the hyphal phenotype may not be an absolute requirement for the physical association of *P. aeruginosa* to fungal cells [14,17].

### 2.2. Role of Quorum Sensing Molecules in the Interaction

*P. aeruginosa* secretes several types of molecules that may influence the growth of *Candida* spp. [18]. An important category of secreted molecules is quorum sensing molecules (QSM), produced by both organisms [19]. Gram-negative bacteria use N-acyl homoserine lactones (AHL) as QSM. At high population densities, when the concentration of AHL is above the threshold level, it can bind to and activate transcriptional activators to induce expression of target genes [20]. *P. aeruginosa* has two AHL-dependent QS systems, *las* and *rhl*. The LasI autoinducer synthase controls the production of the autoinducer, 3-oxododecanoyl-l-homoserine lactone (3-oxo-HSL), while production of butanoyl homoserine lactone is regulated by RhlI autoinducer synthase. These autoinducers act on their respective transcriptional activators, LasR and RhlR [21,22] and, depending on the culture conditions, may regulate up to 10% of the *P. aeruginosa* genome [23,24]. 3-oxododecanoyl-l-homoserine lactone (3-oxo-HSL), which is required for the production of surface adherence proteins on *P. aeruginosa* cells, is also important for adherence of *P. aeruginosa* to *C. albicans* hyphae [25]. In addition, 3-oxo-HSL could inhibit the yeast to hyphal switch of *C. albicans* in a dose-dependent manner [26,27]. Interestingly, Trejo-Hernández and co-workers (2014) found that hypoxia influences the ability of *P. aeruginosa* to inhibit *C. albicans* filamentation, due to a decrease in AHL production [28]. Another QS signal, 2-heptyl-3-hydroxyl-4-quinolone signal or *Pseudomonas* quinolone signal (PQS), which is released by *P. aeruginosa* during late exponential phase [29,30], is induced by the LasI/R system and inhibited by the RhlI/R system [31]. This signal not only modulates swarming motility of *P. aeruginosa* [32,33], but also induces the production of several virulence factors, including phenazines [34]. *Pseudomonas* quinolone signal and its precursor, 2-heptyl-4-quinolone, represses *C. albicans* biofilm formation [35]. These combined effects may lead to the dispersal of *C. albicans* cells in the presence of *P. aeruginosa* [25,36].

Lee and co-workers (2013) provided evidence of an additional QS network, integrated QS system or IQS, that is capable of integrating the stress cues from the environment with the QS network [37,38]. Although dependent on environmental factors, this QS system was shown to function below the *las* QS circuit in a hierarchical manner and was capable of regulating virulence factors through induction of the PQS and *rhl* QS systems. The impact of this addition player in the QS regulatory circuit in the interaction with *C. albicans* is unknown.

Recently, Martínez and co-workers (2019) discovered another cell-density dependent system that is reliant on the production of oxidized fatty acids (oxylipins), termed oxylipin-dependent quorum sensing (ODS) [39]. This system affects the motility of *P. aeruginosa*, but functions independently of the hierarchical circuit of *P. aeruginosa*. The activity of this system was induced with the addition of the polyunsaturated fatty acid, oleic acid, that may be available during infection by this pathogen. This raises the question if oxylipins may play a role in the interaction between *P. aeruginosa* and *C. albicans*, as *C. albicans* is also known to produce oxylipins from polyunsaturated fatty acids that can affect morphology in a QS manner [40]. The role of fatty acid metabolites in the interaction is discussed later.

*Candida albicans* also produces QSMs [41]. The QSM, farnesol, inhibits germ tube formation, similar to 3-oxo-HSL. This effect may be due to inhibition of the Ras1p-controlled pathway involved in hyphal growth [36]. Farnesol inhibits *P. aeruginosa* PQS and pyocyanin (PYO) production in a dose dependent manner [19]. It also influences expression of *P. aeruginosa* virulence-related proteins [42], including haemolysin production [43] and inhibits swarming motility [27]. Another *C. albicans* QSM, tyrosol, stimulates hyphal formation [44], especially during early and intermediate stages of biofilm formation [45]. Physiologically relevant concentrations of tyrosol could also inhibit haemolysin production and secretion of protease by clinical isolates of *P. aeruginosa* [43].

### 2.3. Role of Phenazines in the Interaction

Pyocyanin or methyl-1-hydroxyphenazine, is produced by *P. aeruginosa* during the early stationary phase [46,47] and is the best studied bacterial phenazine. It plays a major role in maintaining NADH/NAD+ ratio stability in *P. aeruginosa* cells under oxygen limitation, since *P. aeruginosa* has a limited fermentation capability [48]. It acts as an alternative terminal electron acceptor and decreases the NADH/NAD+ ratio. When oxygen becomes available again, PYO is re-oxidized, possibly leading to the production of reactive oxygen species (ROS). This production of ROS may contribute to the toxic effect of phenazines on eukaryotes, including fungi [49,50]. Pyocyanin also decreases cyclic adenosine monophosphate (cAMP) [51], which explains the observed inhibition of the *C. albicans* yeast to hyphae transition, which requires cAMP. Gibson and co-workers (2009) co-incubated *P. aeruginosa* and *C. albicans* and observed a red pigment inside *C. albicans* cells [52]. The presence of the red pigment was linked to repression of *C. albicans* viability. The precursor of this red pigment was identified as 5-methyl phenazine-1-carboxylic acid (5-MPCA). The related phenazine, phenazine-1-carboximide (PCN), also showed broad-spectrum antifungal activity, including against *C. albicans* and *C. glabrata* and inhibited the yeast to hyphal switch in *C. albicans* [53]. The mechanism of action of PCN was increased ROS production, mitochondrial membrane hyperpolarization, and eventually, apoptosis. Interestingly, Morales and co-workers (2013) found that low concentrations of PYO, phenazine methosulfate, and PCN specifically inhibited respiration, leading to increased fermentation and lower extracellular pH, which inhibited *C. albicans* yeast-to-hyphal switch [54]. Furthermore, Briard and co-workers (2015) indicated that sub-inhibitory concentrations of phenazines were capable of reducing ferric iron, making it more bio-available for co-inhabitants, such as *Aspergillus fumigatus* [55]. This is a phenomenon that may apply to *C. albicans* as well.

Chen and co-workers (2014) studied the effect of the *C. albicans* fermentation product, ethanol, on *P. aeruginosa* [56]. They found that ethanol inhibited swarming motility, but stimulated adhesion, subsequent biofilm formation and production of 5-MPCA and PCN. This implies the existence of a positive feedback loop where *C. albicans*-produced ethanol enhances *P. aeruginosa* biofilm formation and production of antifungal phenazines, which in turn stimulates ethanol production in *C. albicans*. Several other phenazines (phenazine-1-ol, phenazine-1-carboxylic acid, and PCN) are also inhibitory towards *C. albicans* and *C. tropicalis*, with phenazine-1-carboxylic acid having the lowest minimum inhibitory concentration [57]. These phenazines also have a synergistic effect with azoles against *Candida* species. All of this suggests that the presence of phenazine-producing organisms, such as *P. aeruginosa*, may influence co-infection by *Candida* spp. as well as the treatment of such fungal co-infection.

### 2.4. Role of Fatty Acid Metabolites in the Interaction

As stated above (Section 2.2), *Candida* spp. can convert exogenous (or host-derived) fatty acids into various lipid metabolites, including hydroxy fatty acids and eicosanoids, such as prostaglandins [40,58,59,60,61,62,63,64,65,66,67], that may influence co-infecting microbes [68] or the host [69,70]. One of the best-studied *Candida* lipid metabolites is the eicosanoid, prostaglandin E_2_ (PGE_2_), which has been reported in *C. albicans*, *C. dubliniensis*, *C. glabrata*, *C. parapsilosis* and *C. tropicalis*. *P. aeruginosa* also secretes an array of fatty acid metabolites, including hydroxy fatty acids [71,72,73] as well as PGE_2_ and prostaglandin F_2α_ [74]. Martínez and coworkers (2019) studied the effect of hydroxy fatty acids produced by the action of diol synthase in *P. aeruginosa* and found that the products, 10-hydroxy-octadecenoic acid (10-HOME) and 7,10-dihydroxy-octadecenoic acid (7,10-DiHOME), caused transcriptomic and proteomic alterations in *P. aeruginosa* [39,71]. This reflected in reduced swarming as well as swimming motility in a dose-dependent manner, as well as strongly promoting type IV pilus-driven twitching movement. These metabolites also promoted micro-colony and subsequent biofilm formation in vitro. Interestingly, 7,10-DiHOME has an inhibitory effect on the growth of *C. albicans* [75], while another hydroxy fatty acid, 3-hydroxy tetradecadienoic acid (3-OH 14:2), which is an endogenous linoleic acid metabolite of *C. albicans*, stimulates germ tube formation, even in the presence of farnesol [40]. Fourie and co-workers (2017) investigated the influence of co-incubation of *C. albicans* and *P. aeruginosa* on eicosanoid production from exogenous arachidonic acid by mixed species biofilms [74]. They found that, although co-incubation decreased the ability of individual colony forming units (CFUs) in the biofilm to produce eicosanoids, the final concentrations of all the studied eicosanoids (i.e., PGE_2_, PGF_2α_, and 15-hydroxy eicosatetraenoic acid) increased compared to single species biofilms. This was due to the increase in *P. aeruginosa* CFUs in the mixed species biofilms. Although it is known that PGE_2_ enhances serum-induced germ tube formation in both *C. albicans* and *C. dubliniensis* [60,66,76], the influence of these eicosanoids on *P. aeruginosa* is unknown.

### 2.5. Role of Competition for Iron in the Interaction

Purscke and co-workers (2012) compared the secretome of single and mixed species biofilms of *C. albicans* and *P. aeruginosa* [77]. They found an overall increase in secreted proteins of mixed species biofilms relative to single species biofilms. This increase was due to increased *P. aeruginosa* secreted proteins, many of them related to production of the *P. aeruginosa* siderophore, pyoverdine. The authors speculated that this was caused by competition for iron in multispecies biofilms. Chelation of iron by pyoverdine decreases its availability to *C. albicans*, leading to the observed inhibition of yeast growth. Because pyoverdine production is increased in mixed biofilms, the virulence of *P. aeruginosa* might also be upregulated in mixed biofilms [28]. An interesting observation was made by Nazik and co-workers (2017) who found that *P. aeruginosa* bacteriophages can bind to and inhibit *C. albicans* (and possibly *C. kefyr*) [78]. This inhibition is likely due to the ability of the bacteriophages to bind iron into tight complexes, precipitating it out of the medium. This denies iron to the yeast, inhibiting both biofilm and planktonic cells. A summary of the interaction between *C. albicans* and *P. aeruginosa* is given in Figure 1.

## 3. In Vivo Interactions between *Candida* spp. and *Pseudomonas aeruginosa*

Although studying the interaction between organisms *in vitro* has the advantage of being able to limit the number of variables, this does not always provide a true picture of the potential interactions of co-infecting organisms in a living host with an active immune system. In addition, different body sites provide unique environments in terms of nutrient availability, presence of microbiota and immunity. Therefore, pathogens may exhibit unique patterns of growth and virulence in separate niches of the human body [79]. Due to this, a brief review of the documented interaction between *C. albicans* and *P. aeruginosa* in different body sites will be given.

### 3.1. Interaction in the Lungs

Kerr (1994) was the first to report inhibition of *C. albicans* in the lungs of postoperative patients after subsequent colonization by *P. aeruginosa* [80]. This inhibition was reversed by antibiotic treatment of the *P. aeruginosa* infection. This observation was followed by studies that indicated that prior colonization by *Candida* spp. increases the susceptibility of the host to *P. aeruginosa* infection [81,82,83,84] and that the risk can be reduced by antifungal treatment [85]. This could be explained by Roux and co-workers (2013) who found that the *Candida*-induced Th1-Th17 immune response caused an increase in interferon-gamma (IFN-gamma) [86]. This led to the inhibition of the expression of scavenger receptors on rat alveolar macrophages, and thus to a decrease in *P. aeruginosa* phagocytosis [87]. Interestingly, Chen and co-workers (2014) found that *C. albicans* produced ethanol promotes *P. aeruginosa* colonization of lung epithelial cells, and Greenberg and co-workers (1999) demonstrated that ethanol could inhibit the clearance of *P. aeruginosa* from the lungs in a rat model of infection by inhibiting macrophage recruitment [56,88]. Ader and co-workers (2011) established a murine infection model with short term colonization by *C. albicans* prior to *P. aeruginosa* infection and found that this leads to a reduction in *P. aeruginosa* load compared to infection in the absence of *C. albicans* [89]. Mear and co-workers (2014) found that *C. albicans* induces secretion of interleukin-17 (IL-17) and IL- 22, leading to the production of antimicrobial peptides, as well as the mobilization of phagocytic cells [90]. Bergeron and co-workers (2017) developed a zebra fish swimbladder infection model to study the effect of *C. albicans*-*P. aeruginosa* co-infection on virulence [91]. This model clearly showed a synergistic virulence associated with increased *C. albicans* pathogenesis and inflammation. It is thus clear that the model and the specific methods used for infection influence the outcome of infection and should be taken into account when comparing results regarding the interactions of *Candida* spp. and *P. aeruginosa* in animal models.

Several studies of CF patients have indicated pulmonary co-infection with *C. albicans* and bacteria [92,93,94,95,96]. In this context, the co-infection of *C. albicans* and *P. aeruginosa* has been well documented [97]. Haiko and co-workers (2019) indicated a higher incidence of co-existence of *P. aeruginosa* with *Candida* spp. compared to other respiratory disorders [98]. Although the clinical impact of fungi in lungs is not yet fully understood [99], it is known that the presence of *C. albicans* in CF sputum is a very strong predictor of co-colonization with *Pseudomonas* spp., as well as of hospital-treated exacerbations [100]. However, Haiko and co-workers (2019) provided evidence that *Candida* spp. colonization does not exclusively lead to higher existence of pathogenic bacteria, indicating that this aspect still needs further validation [98]. Co-colonization with *C. albicans* and *P. aeruginosa* also led to a significant decline in lung function compared to patients without *C. albicans*. Kim and co-workers (2015) provided evidence of adaptation of *C. albicans* in the CF lung [101]. They isolated *C. albicans* strains that independently accumulated loss-of-function mutations in the *NRG1* gene, the product of which is a transcription factor that is known to repress filamentation in *C. albicans* [102]. Strikingly, these strains were impervious to the filamentation-suppressing effects of co-culture with *P. aeruginosa* and an analog of PYO, phenazine methosulfate. This study indicated that this adaptation may be common in the CF lung environment [101].

Patients on mechanical ventilation are also particularly susceptible to colonization by *Candida* spp. and subsequent *P. aeruginosa* ventilator-associated pneumonia (VAP) [81]. In 214 patients with *Pseudomonas* spp. VAP included in this study, who also had *Candida* spp. in their respiratory tracts, the most common *Candida* spp. were *C. albicans*, *C. glabrata* and *C. tropicalis*. Hamet and co-workers [83] found that mortality of intensive care patients with VAP was significantly increased by *Candida* spp. co-colonization and that *Candida* colonization was an independent predictor of death. According to Delisle and co-workers (2011), it is unclear whether colonization of the respiratory tract by *Candida* spp. is merely an indication of disease severity or contributes to the observed worse clinical outcomes in these patients [103]. Since respiratory colonization with *Candida* spp. alone can increase levels of tumor necrosis factor alpha and IFN-gamma in the lung [82] and, as stated above, IFN-gamma can impair the function of alveolar macrophages, it may be possible that colonization by *C. albicans* could inhibit an antibacterial immune response. In a recent review by Pendleton and co-workers [104] dealing with the significance of *Candida* in the human respiratory tract, the question regarding the clinical relevance of treating such occurrences with antifungal drugs is raised. Although treatments with antifungals in experimental models have yielded positive results [86,105], this was not always the case in human patients [85,106,107]. Due to these discrepancies in the results, no firm treatment guidelines exist, and clinical practice varies.

### 3.2. Interaction in Wounds

Chronic wounds, such as diabetic foot ulcers, pressure ulcers, and venous leg ulcers are prone to infection by polymicrobial biofilms [108,109]. Many of these are co-infections by *Candida* spp. (*C. parapsilosis*, *C. tropicalis* and *C. albicans*) and bacteria, including *P. aeruginosa* [110]. Despite this, infections in chronic wounds are typically treated with antibiotics, ignoring the fungal component, often leading to unresolved infections. Townsend and co-workers (2017) used a biofilm model, consisting of a complex cellulose matrix, to mimic the wound environment and polymicrobial biofilms consisting of *C. albicans*, *P. aeruginosa* and *Staphylococcus aureus* [109]. By treating the polymicrobial biofilms with antibiotics alone or in combination with fluconazole, they could change the compositions of the biofilms without affecting the overall cell numbers. They also found that the *C. albicans* population is an important driving force within the biofilm and is correlated to the total number of cells in the biofilm. This is due to the physical support and protection provided by the hyphae and extracellular matrix of *C. albicans*.

Another type of wound prone to infection by both *Candida* spp. and *P. aeruginosa* is thermal burn wounds. Although burn wounds are initially sterile, within the first 48 hours they typically become infected by Gram-positive bacteria. During the first week the population shifts to Gram-negative bacteria [111]. Lethal *Candida* infections often follow or occur concomitantly with bacterial infections [112]. The *Candida* spp. identified from burn wounds include *C. albicans*, *C. tropicalis,* and *C. parapsilosis* [113]. Using a burned mouse model to study the effect of a recent *P. aeruginosa* infection on the development of systemic *C. albicans* infection, Neely and co-workers (1986) found that burned mice pre-infected with a sublethal dose of *P. aeruginosa*, followed by a sublethal challenge with *C. albicans*, had a significantly higher mortality rate than either unburned mice, challenged in the same way, or burned mice infected with only one of the microbes [112]. They found that burned mice challenged with both organisms died due to *Candida* infection and indicated the importance of proteolytic enzymes secreted by *P. aeruginosa* in allowing the establishment of lethal *C. albicans* infections. This is in contrast to a study by Gupta and co-workers (2005), who found significant in vivo inhibition of *C. albicans* growth in the presence of *P. aeruginosa* in burn wounds [114].

### 3.3. Interaction in the Gastrointestinal Tract

*C. albicans* is a commensal member of the gastrointestinal microbiota of many mammals, including humans [115,116]. Certain immunocompromised patients, such as cancer patients, are at risk of developing invasive *Candida* infections that may originate from the gastrointestinal system. Similarly, these patients are also at higher risk of *P. aeruginosa* bloodstream infections originating from the gastrointestinal tract [117]. In order to study the interaction between *C. albicans* and *P. aeruginosa*, a neutropenic murine model for gastrointestinal co-colonization was developed by Lopez-Medina and co-workers (2015) [118]. The reported antagonism between *C. albicans* and *P. aeruginosa*, often seen in vitro, was not seen in this model, with levels of both *P. aeruginosa* and *C. albicans* unaffected by co-colonization. In addition, mice co-colonized with *P. aeruginosa* and *C. albicans* had significantly lower mortality compared to mice colonised with only *P. aeruginosa*. All dead mice exhibited evidence of *P. aeruginosa* dissemination. The authors showed that *C. albicans* supressed expression of *P. aeruginosa* pyochelin and pyoverdine genes. Deletion of these genes did not affect the ability of *P. aeruginosa* to colonize the gastrointestinal tract, but did decrease the dissemination and virulence of *P. aeruginosa*. This virulence was restored after oral iron supplementation. This points to the importance of iron in the in vivo interaction between *C. albicans* and *P. aeruginosa*. In addition, using cultured colonocytes, they also found that *C. albicans* secreted proteins inhibited the production of cytotoxic exotoxin A. Lamont and co-workers (2002) indicated that pyoverdine regulates the production of exotoxin A, so the decrease in exotoxin A levels may also be an additional effect of *C. albicans* mediated decrease in pyoverdine production [119]. Regardless of the mechanism behind the reduction in exotoxin A, this may prevent the damage caused by *P. aeruginosa* in the gastrointestinal tract and possibly prevent dissemination. Table 1 summarizes the interaction between *C. albicans* and *P. aeruginosa in vivo*.

## 4. Phenotypic Plasticity in Polymicrobial Interactions

An impressive characteristic of *C. albicans* is its phenotypic plasticity, with this fungus able to exhibit up to nine distinct cell types that are dependent on environmental cues [120]. These include the classical cell types, yeast, hyphae, pseudohyphae, and chlamydospores. Furthermore, in addition to the white phenotype with heterozygosity at the mating type locus (a/α; mostly observed in laboratory conditions and infection), a white phenotype exists with either the a or α genotype, as well as an opaque phenotype with heterozygosity at the mating type locus, a/α, with opaque (homozygous, a or α), grey, and gastrointestinally-induced transition (GUT) phenotypes [120,121,122,123]. These non-classical phenotypes are highly dependent on the expression of the white-opaque regulator, Wor1p, that causes heritable changes, eliciting transcriptional changes, which flow over into virulence. The expression of this regulator is dependent on environmental cues, such as *N*-acetylglucosamine, a component of the bacterial cell wall, CO_2_, and hypoxia—an environmental circumstance frequently found in the gastrointestinal tract [122,124,125]. These phenotypes exhibit altered metabolic specialization and mating potential. For example, expression of GUT cell genes involved in iron uptake and glucose utilization is down-regulated, with an upregulation in genes involved in utilization of *N*-acetylglucosamine and short-chain fatty acids, making these cells well suited for commensal growth with resident bacteria in the gastrointestinal tract [122]. Furthermore, whereas white cells have an increased expression of genes involved in pathways associated with fermentative growth, opaque cells are more primed for oxidative respiration [121,123]. Furthermore, Malavia and co-workers (2017) described a hyper-adherent goliath phenotype induced by zinc limitation [126].

The influence of some of the classical phenotypes, such as yeast and hyphae, on the interaction with *P. aeruginosa* has been evaluated. Less information regarding the influence of non-classical phenotypic alterations on the interaction between *C. albicans* and pathobionts such as *P. aeruginosa* is available. This is especially interesting as certain niches, where co-occurrence of *C. albicans* and *P. aeruginosa* has been observed, may have the presence of environmental cues that could potentially elicit a partial or full switch to the mentioned opaque or GUT phenotypes in *C. albicans*, such as in the CF lung where bacterial cell wall components such as *N*-acetylglucosamine, are abundant due to the presence of bacteria [102], and hypoxic conditions are present [127]. However, further research is required to determine if *P. aeruginosa* could encounter other phenotypes of *C. albicans* besides the classical cell types, and whether this would cause any alterations in the interaction with this bacterium.

Similar to the functional specialization of *C. albicans*, is the remarkable phenotypic variation exhibited by *P. aeruginosa,* driven by hypermutable strains and richness in oxidative and nitrosative stresses, when in the environment of the CF lung [128,129,130]. Genetic and functional phenotypic variation lead to *P. aeruginosa* isolates exhibiting differences in colony morphology, extracellular polysaccharide production, motility, quorum sensing, protease activity, auxotrophy, siderophore production, and growth profiles [131]. These different phenotypes are found together, even in a single patient, and comprise a highly dynamic population of *P. aeruginosa* in chronic CF lung infections due to spatial heterogeneity and niche partitioning [131,132]. In addition, large variation in susceptibility to antimicrobials is seen with little association to morphotype. How these different phenotypes may influence the interaction of *P. aeruginosa* with pathobionts in the CF lung such as *C. albicans* has received little attention. Of special interest is the ethanol production by *C. albicans* that may be able to promote a mucoid phenotype of *P. aeruginosa* in the CF lung [56,133]. This phenotype, associated with an increase in alginate production, a component of the extracellular matrix of *P. aeruginosa*, is associated with long term chronic infection of the CF lung by *P. aeruginosa*. In addition to overproduction of alginate, the adaptation of *P. aeruginosa* to long term chronic CF infection also includes decreased expression and loss of flagellin, the type III secretion system, as well as changes in LPS structure that aid in immune evasion [128,134,135,136,137]. The possible influence of this adaptation of *P. aeruginosa* on *C. albicans* deserves further attention.

## 5. Conclusions and Future Perspectives

The interaction between *C. albicans* and *P. aeruginosa* is frequently used as a model to study the complex interplay between cross-kingdom co-infectious agents. This interaction is often characterized as antagonistic in vitro, with suppression of fungal growth by both physical association as well as secreted factors. Notably, QSMs produced by both species alter the phenotype of the other during association. The discovery of an additional *P. aeruginosa* QS participant in the *las*, *rhl,* and PQS systems, namely IQS, still warrants investigation into the influence during polymicrobial growth with *C. albicans*. Additionally, the identification of the QS activity of oxylipins and eicosanoid production in *P. aeruginosa*, as well as the known production of oxylipins and eicosanoids by *C. albicans*, further motivates research into their effects on fungal-bacterial interactions.

The in vitro antagonism of the interaction between *C. albicans* and *P. aeruginosa* may be a simplistic view of an interaction in which both participants influence the outcome of infection. In addition, this interaction is much more complex in vivo, where either species can alter colonization by the other, promoting or inhibiting disease. This seems to be highly dependent on the site of infection, immune functionality, and prior antimicrobial treatment. Both pathogens exhibit complex regulatory systems that facilitate adaptability to multiple niches within the human body. Due to this, they are rarely isolated in only one niche, as described above in the various infection sites, and exhibit alterations in their phenotypes. The occurrence of the functional specialization of both *C. albicans* and *P. aeruginosa* leading to altered phenotypes also prompts the investigation into the occurrence of these phenotypes in the presence of other microbial species, i.e., *C. albicans* in the presence of *P. aeruginosa,* and vice versa. Additionally, the influence of these phenotypes on the interaction of these two species still requires investigation.

## Figures and Tables

**Figure 1 jof-05-00034-f001:**
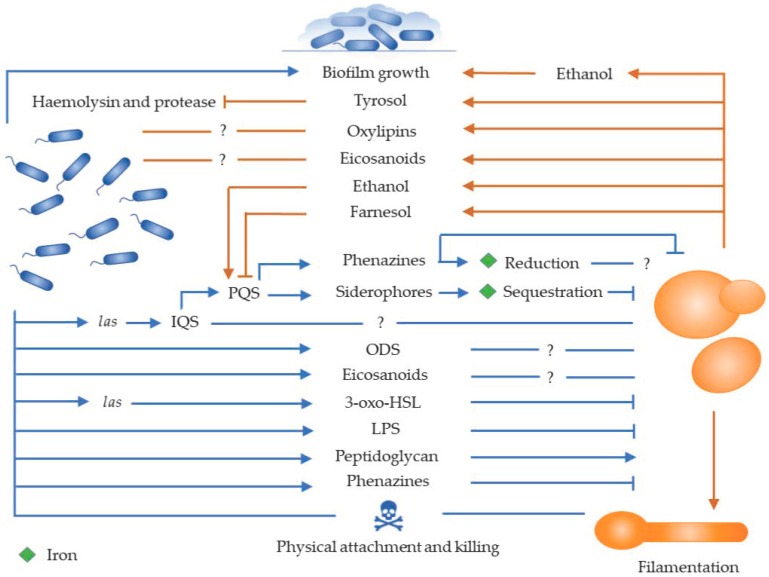
Schematic of the interaction between *Candida albicans* and *Pseudomonas aeruginosa in vitro*. *P. aeruginosa* kills *C. albicans* hyphae through physical attachment. In addition, cell wall components such as lipopolysaccharide (LPS) and peptidoglycan affect filamentation. Furthermore, secreted factors such as phenazines and siderophores, as well as quorum sensing systems, such as *las* (controls the production of autoinducer 3-oxododecanoyl-l-homoserine lactone or 3-oxo-HSL), *Pseudomomas* quinolone signal (PQS), integrated quorum sensing (IQS), and oxylipin dependent sensing (ODS) may affect *C. albicans*. Lastly, *C. albicans* secreted factors such as farnesol, ethanol, oxylipins, and eicosanoids may affect *P. aeruginosa* growth characteristics and biofilm formation. Question mark (?) indicate unknown roles in the interaction.

**Table 1 jof-05-00034-t001:** Summary of the interaction between *Candida* spp. and *Pseudomonas aeruginosa* in vivo in various infection sites.

Infection Site	Type of Study/Model	Observations	Effect of *P. aeruginosa* on *Candida*	Effect of *Candida* on *P. aeruginosa*	Number(s) in Reference List
Lungs	Postoperative monitoring of surgery patients	Inhibition of *C. albicans* after subsequent colonization with *P. aeruginosa*. Reversed with antibiotic treatment	Inhibited	-	[80]
Lungs	Monitoring of patients with mechanical ventilation	Colonization with *Candida* spp. associated with increased risk of *P. aeruginosa* VAP^1^	-	Promoted	[81]
Lungs	Monitoring of patients with VAP ^1^	Increase in isolation of multidrug resistant bacteria such as *P. aeruginosa* when *Candida* spp. are present. Reduced risk for *P. aeruginosa* VAP ^1^ with antifungal treatment	-	Promoted	[83,85]
Lungs	Analysis of sputum samples from CF ^2^ patients	Higher incidence of co-existence between *P. aeruginosa* and *Candida* spp. in CF^2^ patients compared to other respiratory disorders	-	-	[98,100]
Lungs	Analysis of sputum samples from CF ^2^ patients	Presence of hyperfilamentous *C. albicans* with loss of function mutation in *NGR1* in presence of *P. aeruginosa*	No inhibition of filamentation	_	[101]
Lungs	Wistar rat model	Fungal colonization promoted pneumonia by *P. aeruginosa* Reversed with antifungal treatment	-	Promoted	[82,86]
Lungs	Mouse model	Prior *C. albicans* colonization promoted *P. aeruginosa* clearance	-	Clearance enhanced	[89,90]
Mucosa	Zebrafish swimbladder model	Synergistic increase in virulence in co-infection compared to single species infection	Increased virulence	Increased virulence	[91]
Wounds	Microbial populations of deep tissue wounds of patients with type 2 diabetes cultured	*C. albicans* not found in combination with *P. aeruginosa*	Inhibited	-	[106]
Wounds	Mouse model	Pre-infection with *P. aeruginosa* predisposed mice to lethal *C. albicans* infection	*P. aeruginosa* proteolytic enzymes promote lethal *C. albicans* infection	-	[108]
Wounds	Microbial analysis of wounds of burn patients	Inhibition of *Candida* spp. when *Pseudomonas* spp. were present	Inhibited	-	[110]
Wound model	*In vitro* biofilm model to mimic wounds	Biofilms of *C. albicans*, *Staphylococcus aureus* and *P. aeruginosa* created. Monotreatment with antifungal/antibiotic only shifted population dynamics without affecting overall biofilm bioburden		Provides physical support and protection to bacteria	[105]
Gastro-intestinal tract	Neutropenic mouse model	Levels of both *P. aeruginosa* and *C. albicans* unaffected by co-incubation		Decrease in siderophore production and virulence	[114]

^1^ VAP – Ventilator-associated pneumonia; ^2^ CF – Cystic fibrosis.

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
