# Peer review of "Beyond Antagonism: The Interaction Between Candida Species and Pseudomonas aeruginosa"

_jof, 2019, doi:10.3390/jof5020034_

Round 1

Reviewer 1 Report

This review is well conceived and describes an important and interesting area of polymicrobial research that is likely to appeal to a broad audience. While concise, the topic has been competently handled, brought up to date, and is supported by appropriate citations. As such, the only comment would be for a greater degree of proofreading, as in its current form the manuscript contains some minor grammatical/typographical errors that should be rectified.

Author Response

We thank the reviewer for  the kind words. The manuscript has been read for errors and corrected as requested.

Reviewer 2 Report

The review describes in detail the complex interactions between the opportunistic pathogenic fungus C. albicans and the bacterium P. aeruginosa in the context of microbial competition and disease. The authors did a fantastic job describing and summarizing the extensive literature on the subject.

Comments:

The manuscript would benefit from a figure like the one included, but focusing on the interactions in vivo. Alternatively, a table describing the studies in vivo and their conclusions (i.e. antagonistic/synergistic) would also suffice and enhance the reader experience.

The in vivo section would also benefit from data on patient outcomes during these infections and current treatment options.

A section on the host immune response would also enhance the manuscript.

The focus of the discussion should be on the overall interactions described throughout and should serve to identify/describe potential gaps in knowledge in the field as well as provide a possible working model on these interactions. As it stands, it reads as a section of the review focusing on phenotypic plasticity of these organisms and not a discussion section. The authors should rewrite this section. I am happy with this section, but not as a discussion.

Minor comments:

The authors alternate between C. albicans and Candida albicans throughout the manuscript. The same goes for P. aeruginosa. Once they use the full genus and species name, they should use C. albicans or P. aeruginosa throughout the rest of the manuscript.

Author Response

The manuscript would benefit from a figure like the one included, but focusing on the interactions in vivo. Alternatively, a table describing the studies in vivo and their conclusions (i.e. antagonistic/synergistic) would also suffice and enhance the reader experience.

A table summarising the in vivo interaction was included (Table 1, page 8 and page 9).

The in vivo section would also benefit from data on patient outcomes during these infections and current treatment options.

To address this, line 258-260 was included, as well as line 266-271. However, little information is available regarding an overall effect on patient outcome and treatment options due to contrasting evidence. For example, as stated in line 260-263, “it is unclear whether colonisation of the respiratory tract by Candida spp. is merely an indication of disease severity or contributes to the observed worse clinical outcomes in these patients”.

A section on the host immune response would also enhance the manuscript.

As the immune response seems to be dependent on the site of infection, the site-specific host immune responses were included under the headings that describe the site of infection. For example, the host immune response in the lungs are briefly explained in line 218-220 as well as 228-230 and 261-264.

The focus of the discussion should be on the overall interactions described throughout and should serve to identify/describe potential gaps in knowledge in the field as well as provide a possible working model on these interactions. As it stands, it reads as a section of the review focusing on phenotypic plasticity of these organisms and not a discussion section. The authors should rewrite this section. I am happy with this section, but not as a discussion.

The section regarding phenotypic plasticity was moved from the Conclusions and Future Prospectives and placed before this section. Furthermore, the discussion in the Conclusions and Future Prospectives section was expanded.

The authors alternate between C. albicans and Candida albicans throughout the manuscript. The same goes for P. aeruginosa. Once they use the full genus and species name, they should use C. albicans or P. aeruginosa throughout the rest of the manuscript.

This has been corrected as suggested